# Prevalence and Characteristics of Human Papillomavirus Infection in Oropharyngeal Squamous Cell Papilloma

**DOI:** 10.3390/cancers15030810

**Published:** 2023-01-28

**Authors:** Dongbin Ahn, Ji-Hye Kwak, Gil-Joon Lee, Jin-Ho Sohn

**Affiliations:** Department of Otolaryngology-Head and Neck Surgery, School of Medicine, Kyungpook National University, Daegu 41944, Republic of Korea

**Keywords:** squamous papilloma, human papillomavirus, prevalence, genotype

## Abstract

**Simple Summary:**

Studies on human papillomavirus (HPV) infection in oropharyngeal squamous papilloma (OPSP) are lacking, although HPV is a critical oncogenic virus for the development of oropharyngeal cancer. This study evaluated the prevalence and characteristics of HPV infection in OPSP and showed a 14.5% overall prevalence. High-risk HPV accounts for 75% of all HPV infections, with HPV16 being the most prevalent genotype, accounting for 58.3% of all HPV infections. There was a trend toward a higher prevalence of high-risk HPV infection in patients with OPSP aged ≤45 years, never-smokers, and those with multifocal diseases, which corresponded with the clinicodemographic profiles of HPV-associated oropharyngeal cancer. These findings could enhance our understanding of HPV infection in OPSP and could be used as valuable epidemiological data for the management of HPV-associated OPSP.

**Abstract:**

Studies on human papillomavirus (HPV) infection in oropharyngeal squamous papilloma (OPSP) are lacking, although HPV infection has been recognized as the primary cause of oropharyngeal cancer for several decades. This study aimed to evaluate the prevalence and characteristics of HPV infections in patients with OPSP. We retrospectively enrolled patients with histologically confirmed OPSP in whom the presence of HPV infections and p16 expression were evaluated. The results of HPV infection in OPSP were analyzed according to the clinicodemographic profiles. Of the 83 patients included in this study, HPV test results were positive in 12 patients, with an overall prevalence of 14.5%. HPV genotypes involved low-risk and high-risk HPV types in three (3.6%) and nine (10.8%) patients, respectively. The most prevalent genotype was HPV16, accounting for 58.3% of all HPV infections. None of the OPSPs showed p16 IHC positivity. There were trends toward a higher prevalence of high-risk HPV infection in patients with OPSP aged ≤45 years, never-smokers, and those with multifocal diseases. These findings could enhance our understanding of HPV infection in OPSP and be used as valuable epidemiological data for the management of HPV-associated OPSP and regarding the possible efficacy of HPV vaccinations in OPSP.

## 1. Introduction

Head and neck squamous papilloma (SP) is a benign tumor histologically characterized by papillary epithelial growth with fibrovascular cores. It can occur in any part of the head and neck with squamous epithelium [1,2]. Although the estimated annual incidence of head and neck SP has been reported to be 0.5–4.3 per 100,000 individuals depending on the anatomical sites, it is being increasingly diagnosed owing to incidental detection during routine health examinations, particularly the SPs of the pharynx and larynx, which are usually found during endoscopic examination [1,3,4].

Although SP is a commonly encountered disease entity in actual clinical practice, its etiology is still not universally confirmed. Mechanical and chemical irritations have been considered the main causes of conventional SP [1,2]. Since the 1980s, human papillomavirus (HPV) infection has been suggested as the primary cause of SP. Among the head and neck SPs, most studies on HPV infection involve the paranasal sinus, larynx, and oral cavity, possibly due to the more aggressive behavior of SP in the paranasal sinus and larynx or the ease of identification in the oral cavity [1,5,6,7,8]. According to a systematic review in 2013, 23.3% (330/1416) of sinonasal inverted papillomas tested positive for HPV, varying from 0% to 62% in different studies [5]. A recent study reported 38.8% HPV positivity in inverted papilloma, with HPV6, HPV16, and HPV11 being the most prevalent genotypes [9]. For laryngeal SP, the proportion of HPV positivity has been reported to be up to 90–95% in recurrent respiratory papillomatosis, with HPV6 and HPV11 being the two most prevalent genotypes [7,8]. For oral SP, two recent studies involving ≥30 cases reported 48.3% (15/31) and 27.3% (9/33) HPV positivity with HPV6, HPV11, and HPV16 genotypes [10,11].

Though there are sufficient data on HPV infection in SPs of the paranasal sinus, larynx, and oral cavity, there are only a few studies on HPV infection in oropharyngeal SP (OPSP) with various geographic distribution. In two studies conducted in Japan involving 12 and 15 patients with OPSP each, HPV infection was not detected in any of the patients [12,13]. In contrast, in a study conducted in Italy, 3 out of 43 patients with OPSP were positive for HPV infection, with a 6.9% prevalence [10]. In a study involving 63 patients with OPSP in the US, HPV infection was detected in four patients, with a 6.3% prevalence [14]. In a study conducted in Poland, 28 out of 52 (53.8%) patients with OPSP were HPV positive—this is the highest reported prevalence of HPV infection in OPSP thus far [15]. With respect to the treatment of OPSP, there is no consensus on the management of incidentally found OPSP, and observation without treatment has been frequently used in clinical practice. Considering that high-risk HPVs, particularly HPV16, have been recognized as critical oncogenic viruses for the development of oropharyngeal squamous cell carcinoma (OPSCC), and the incidence of HPV-associated OPSCC has been increasing for several decades, more epidemiological and clinical information about HPV infection in OPSP should be obtained to establish appropriate management strategies and to predict the prognosis [16,17,18].

This study aimed to evaluate the prevalence and characteristics of HPV infection in OPSP to enhance our understanding of HPV infection as a causative factor of OPSP, ultimately providing clinically available epidemiological data for the management of HPV-associated OPSP.

## 2. Materials and Methods

### 2.1. Patients

This was a retrospective cohort study conducted between April 2016 and June 2022. The study protocol was approved by our institutional review board, and the need for informed consent was waived owing to the retrospective nature of the study.

The inclusion criteria were patients with SP, (1) which was pathologically confirmed after surgical treatment, and (2) that arose from the oropharynx. The exclusion criteria were patients (1) in whom evaluation of HPV status was unavailable, (2) who had a history of HPV vaccination, (3) who had a history of head and neck cancer, and (4) whose medical records were incomplete on review.

### 2.2. Sample Size Verificaiton

The prevalence of HPV infection in OPSP has been reported to be 0–53.8% in six different studies [10,12,13,14,15,19]. From these six studies, we extracted a total of 197 patients with OPSP and identified 39 HPV-positive cases among them. Thus, the calculated prevalence of HPV infection in OPSP was 19.8%. In addition to the OPSP data, we also referred to the prevalence of HPV infection in overall head and neck SPs to estimate appropriate sample size as sufficient data were lacking on the prevalence of HPV infection in OPSP alone. The prevalence of HPV infection in overall head and neck SPs is reported to be approximately 30%, with 80–90% being laryngeal SPs, 20–40% being sinonasal SPs, and 20–50% being oral SPs [1,5,6,7,8,9,11,14]. Based on these literature reviews, we defined the expected prevalence of HPV infection in OPSP as 20%, precision as 10% (half of the expected prevalence), and the level of confidence interval as 95% for the sample size calculation of the present study [20]. With these parameters, the calculated sample size was 62.

### 2.3. HPV Detection and Genotyping

Detection and genotyping of HPV infection was performed using a CLART HPV2 kit (Genomica, Madrid, Spain) with formalin-fixed paraffin-embedded (FFPE) tissue samples of diagnosed OPSP [21]. CLART HPV2 allows the detection of 35 different HPV genotypes, including 18 high-risk (16, 18, 26, 31, 33, 35, 39, 45, 51, 52, 53, 56, 58, 59, 66, 68, 73, and 82) and 17 low-risk (6, 11, 40, 42, 43, 44, 54, 61, 62, 70, 71, 72, 81, 83, 84, 85, and 89) genotypes. Two internal controls, a DNA control for sample sufficiency and an amplification control for each reaction, were used for the assay. The detection of HPV was achieved by amplification of a 450 bp fragment of the HPV L1 locus. Subsequently, visualization and genotyping was performed using a CLART microarray platform and type-specific probe (Figure 1) [22]. The CLART HPV2 assay demonstrated good performance comparable to that of real-time polymerase chain reaction based on the reference protocol, and the European Union approved in vitro diagnostic use of this assay for cytological and archival FFPE tissues [23].

### 2.4. Histopathological Examination and P16 Immunohistochemistry

OPSP was diagnosed histopathologically by a specialized head and neck pathologist. Along with routine histopathological examination using hematoxylin and eosin staining, p16 expression reflecting the biological activity of HPV infection was analyzed using immunohistochemistry with CINtec^®^ p16 Histology (Ventana, Medical System, Tucson, AZ, USA) in all OPSPs. The intensity of staining was scored using a known p16 expression in HPV-positive OPSCC as a positive control. Overexpression of p16 was defined as homogeneous, strong nuclear, and cytoplasmic staining present in >70% of tumor cells [24]. All other staining patterns were considered negative.

### 2.5. Assessment Parameters and Statistical Analysis

The primary outcome variables were defined as the overall prevalence and genotype of HPV infection in OPSPs. The secondary outcome variable was the prevalence of HPV infection as a function of clinicodemographic characteristics, such as age, sex, smoking status, tumor focality, and anatomical subsites of the oropharynx. The presence of dysplasia on histopathological examination, p16 IHC results, and recurrence after treatment were also evaluated.

Statistical analyses were performed using SPSS (version 18.0; SPSS Inc., Chicago, IL, USA). Continuous data are presented as the mean ± standard deviation. Categorical variables are presented as number of patients (%). The associations between HPV infection and clinicopathological parameters were assessed using an independent two-sample t-test for continuous variables and the chi-square test or Fisher’s exact test for categorical variables. Statistical significance was defined as a two-sided *p*-value < 0.05.

## 3. Results

### 3.1. Baseline Patient Characteristics

Figure 2 shows the flowchart of patient enrollment. Between April 2016 and July 2022, 83 patients with OPSP met our inclusion criteria and were enrolled in this study. The mean age was 51 ± 16.7 years, and 54 (65.1%) and 29 (34.9%) patients were men and women, respectively (Table 1). OPSP was found incidentally without clinical signs or symptoms in 60 (72.3%) patients, while 21 (25.3%) and two (2.4%) patients had foreign body sensation and chronic cough as clinical presentations, respectively. The anatomical subsites of OPSP involved the tonsil, tongue base, soft palate, and posterior wall in 36 (43.4%), 18 (21.7%), 28 (33.7%), and 6 (7.2%) patients, respectively. OPSP was unifocal and multifocal in 79 (95.2%) and 4 (4.8%) patients, respectively. None of the OPSPs showed either dysplastic changes upon pathological examination or recurrence after surgical treatment.

### 3.2. Overall Prevalence of HPV Infection and Its Genotypes

The HPV test was positive in 12 patients; thus, the overall prevalence of HPV infection in OPSP was 14.5% (Table 2). HPV genotypes included low-risk and high-risk types in three (3.6%) and nine (10.8%) patients, respectively. The most prevalent type of HPV infection was HPV16, which was detected in 7 (58.3%) of 12 patients. Thus, the prevalence of HPV16 infection in patients with OPSP was 8.4%. Among the high-risk types, HPV58 and coinfection with HPV39 and HPV66 were also identified in one patient each. Among the low-risk types, HPV11 and HPV84 were identified in two and one patients, respectively. IHC for p16 was negative in all 83 patients, regardless of the HPV infection status. The detailed characteristics of the 12 patients with HPV infection in the OPSP group are presented in Table 3.

### 3.3. Prevalence of HPV Infection According to Clinicodemographic Profiles

The prevalence of overall HPV infection in OPSP were 14.8% and 14.3% in patients aged ≤45 and >45 years, respectively (*p* = 0.595); 13.0% and 17.2% in men and women, respectively (*p* = 0.412); 16.3% and 12.5% in never-smoker and ex- or current-smoker, respectively (*p* = 0.431); and 13.9% and 25.0% in unifocal and multifocal disease, respectively (*p* = 0.471). The prevalence of HPV infection according to oropharyngeal subsites, including the tonsil, tongue base, soft palate, and posterior wall, was 13.9%, 11.1%, 10.7%, and 33.3%, respectively (*p* = 0.517) (Table 4).

The prevalence of high-risk HPV infection in OPSP was 14.8% and 8.9% in patients aged ≤45 and >45 years, respectively (*p* = 0.324); 9.3% and 13.8% in men and women, respectively (*p* = 0.386); 14.0% and 7.5% in never-smokers and ex- and current-smokers, respectively (*p* = 0.279); and 10.1% and 25.0% in unifocal and multifocal disease, respectively (*p* = 0.374). The prevalence of high-risk HPV infection according to oropharyngeal subsites, including the tonsil, tongue base, soft palate, and posterior wall, was 11.1%, 11.1%, 7.1%, and 16.7%, respectively (*p* = 0.895).

The proportion of high-risk genotypes in all HPV infections was 100.0% and 62.5% in patients aged ≤45 and >45 years, respectively (*p* = 0.255); 71.4% and 80.0% in men and women, respectively (*p* = 0.636); 85.7% and 60.0% in never-smokers and ex- or current-smokers, respectively (*p* = 0.364); and 72.7% and 100.0% in unifocal and multifocal disease, respectively (*p* = 0.750). The proportion of high-risk genotypes in all HPV infections according to oropharyngeal subsites including the tonsil, tongue base, soft palate, and posterior wall was 80.0%, 100.0%, 66.7%, and 50.0%, respectively (*p* = 0.895).

## 4. Discussion

This study showed that the prevalence of overall HPV infection in OPSP was 14.5% (12/83), with HPV11, HPV16, HPV58, HPV39/66, and HPV84 genotypes. Data on HPV infection in OPSP are scarce because there have been no studies focusing on HPV infection of OPSP alone, but a few studies have included the oropharynx as an anatomical site for the evaluation of HPV infection in overall head and neck SPs [1,10,12,14,15]. In previous studies, the prevalence of HPV infection in OPSP varied from 0% to 53.8% in different countries (Japan, Italy, US, and Poland), and this wide variation might be attributed to the small number of cases and a large variety of HPV detection methods, along with rapid advancement in HPV detection techniques over time [10,12,14,15]. In the two most recent studies involving ≥40 cases of OPSP in 2017 and 2020, the prevalence of HPV infection was 6.9% (3/43) and 6.3% (4/63), respectively [10,14]. Thus, our study showed a relatively higher prevalence of HPV infection (14.5%) than those reported in recent studies; however, it was within the range of previously reported prevalence from various geographic regions [10,12,13,14,15,19]. Indeed, we believe that the result of our study is close to the true prevalence of HPV infection in OPSP by including the largest number of cases to date and minimizing study biases in patient enrollment and HPV testing.

Given the lack of sufficient studies on the prevalence of HPV infection, there is limited data on HPV genotypes in OPSP with large heterogeneity, even among the results of previous studies. Since 2010, only four studies have addressed specific HPV genotypes in OPSP, and HPV6, HPV11, HPV51, and HPV74 have been identified in different studies, showing a high predominance of low-risk HPV [1,10,14,15]. A study in 2017 reported HPV11 and HPV6 as the two most prevalent genotypes, with a prevalence of 53.8% (28/52) [15]. However, this study used an HPV test kit only for HPV6 and HPV11 among low-risk HPVs, and high-risk HPVs were not detected at all in their series. Another study on 43 OPSPs in 2017 reported three cases of HPV infection with genotypes HPV6, HPV51, and HPV 74 [10]. In contrast to previous studies, in our study, the most prevalent genotype of HPV infection was HPV16, which accounted for 58.3% (7/12) of all HPV infections, showing a high prevalence of high-risk HPV (9/12, 75%). The results of HPV genotypes in OPSP are quite different from those of other head and neck SPs, including the paranasal sinus, larynx, and oral cavity, where the most prevalent HPV genotypes have been reported as HPV6 and HPV11, the low-risk HPVs [1,5,6,7,8,9,23]. However, our results are concordant with those of studies on HPV infection of the oropharynx, particularly tonsils, in tumor-free patients. A systematic review of oral and oropharyngeal HPV infections involving 4581 cancer-free subjects demonstrated that the prevalence of HPV infection was 4.5%, with HPV16 being the most prevalent, accounting for 28% of all HPVs detected [25]. In a recent systematic review of tonsillar HPV infection in healthy tumor-free patients, HPV16 was the most prevalent genotype, accounting for 78.6% (16/28) of all HPV detected, although various HPV genotypes including HPV6, HPV11, HPV16, and HPV31 were identified [26]. Moreover, our previous study on tonsillar HPV infection in 362 tumor-free patients also indicated HPV16 as the most prevalent genotype, seen in six (75%) of eight patients with tonsillar HPV infection [21]. HPV16 is also the predominant genotype in HPV-associated OPSCCs, accounting for more than 80% of HPV infections [16]. Therefore, considering the strong oropharynx-tropic inclination of HPV16 infection in both normal and tumorous oropharyngeal tissues, the results that show HPV16 as the most prevalent genotype in OPSP are not outrageous, but distinguish the reliability of the present study from previous studies [17].

In the present study, there were no differences in the prevalence of overall HPV infection according to clinicodemographical profiles, such as age, sex, smoking, tumor focality, and anatomical subsites. However, there were trends toward a higher prevalence of high-risk HPV infection in patients with OPSP aged ≤45 years, never-smokers, and multifocal diseases. Interestingly, these findings were consistent with the clinicodemographical profiles of HPV-associated oropharyngeal cancer, which include younger age, absence of traditional risk factors, such as alcohol and tobacco, and frequent metastatic disease [17,18]. This clinicodemographic similarity between high-risk HPV-associated OPSP and OPSCC may reflect the biological characteristics of high-risk HPV, particularly HPV16, the most prevalent type of HPV infection in both OPSP and OPSCC [17]. However, we cannot estimate the possible clinical implications of high-risk HPV-associated OPSP in the transformation into HPV-associated OPSCC because p16 overexpression, which acts as a surrogate marker of the presence of biologically active HPV, was not found even in high-risk HPV-associated OPSP, suggesting that none of them had transcriptionally active HPV infection that could lead to HPV-driven carcinogenesis [21]. It is well documented that high-risk HPV infection does not always induce carcinogenesis; however, it could progress toward several different biological and clinical pathways. HPV infection would be eliminated spontaneously in most patients, may be associated with the development of benign tumors in some patients (HPV-associated OPSP and genital warts), or may induce HPV-driven carcinogenesis in few patients (HPV-associated cervical cancer and oropharyngeal cancer) [22,27,28,29,30]. Indeed, HPV infection is only the first step in the HPV-driven carcinogenesis, and many additional genetic and signaling alterations involving p53 and Rb degradation by E6 and E7, respectively, as well as PI3K/AKT/mTOR signaling, are required for HPV infection to ultimately lead to HPV-associated cancer [17,31,32,33]. Therefore, HPV-associated OPSP with negative p16 was considered another phenotype of oropharyngeal HPV infection that might present with a different biomolecular pathogenesis compared to that of HPV-driven carcinogenesis. Nevertheless, we consider complete surgical resection of OPSP a better therapeutic approach than observation without treatment to obtain its specific histological and virological information.

Recently, there has been growing interest and discussion about the effects of HPV vaccination to prevent and treat HPV infection or HPV-related diseases of the head and neck. In a cross-sectional study of the US population aged 18 to 33 years (N = 2627) within the National Health and Nutrition Examination Survey 2011 to 2014, HPV vaccination was associated with reduced vaccine-type oral HPV infection (0.1% in vaccinated individuals vs. 1.6% in unvaccinated individuals; *p* = 0.008) [34]. Furthermore, a recent systematic review of the therapeutic effect of the HPV vaccine involving 153 cases of recurrent respiratory papillomatosis reported that 80.4% of patients experienced either complete or partial remission of their papilloma, a decrease in surgical intervention, or an increase in time between surgical interventions [35]. Although no epidemiological studies on the preventive effect of HPV vaccination on head and neck SP are currently available, there is no reason to believe that the molecular mechanism underlying vaccine efficacy in the head and neck would differ from that in the anogenital tract, where a promising effect of the HPV vaccine has already been demonstrated [36,37]. Currently, three FDA-approved HPV vaccines are available as follows: (1) Gardasil (Merck & Co., Whitehouse Station, NJ, USA), a quadrivalent vaccine targeting HPV6, HPV11, HPV16, HPV18, and HPV; (2) Cervarix (GlaxoSmithKline, Brentford, London, UK), a bivalent vaccine targeting HPV16 and HPV18; and (3) Gardasil 9 (Merck & Co, Whitehouse Station, NJ, USA), a nonavalent vaccine targeting HPV6, HPV11, HPV16, HPV18, HPV31, HPV33, HPV45, HPV52, and HPV58 [35,36,37]. Given that HPV genotypes detected in the present study were HPV11, HPV16, HPV58, HPV39/66, and HPV84 and its proportion in all HPV infections, a nonvalent vaccine would be the most effective among these three types of HPV vaccines, covering 83.3% (10/12) of HPV infections in HPV-associated OPSP. Therefore, it is expected that the incidence of OPSP will decrease by up to 80% in the future if HPV vaccination is included in national vaccination programs and gender-neutral vaccination becomes more popular for achieving herd immunity against HPV infections. However, this study excluded patients who received HPV vaccination in order to identify the true prevalence of HPV infection in OPSP; thus, we could not estimate the effect of vaccination on HPV infection in OPSP. In the future, a population-based epidemiological study including both vaccinated and unvaccinated patients may provide information on the true effect of HPV vaccination on the development of OPSP as well as the characteristics of HPV infection as a function of HPV vaccination.

This study had several limitations. First, although this is the largest study on HPV infection in OPSP, the number of study subjects was still too small to represent the true prevalence of HPV infection in OPSP and to demonstrate statistical differences as a function of the clinicodemographic characteristics. Second, given that the prevalence and characteristics of HPV infection are highly affected by sexual and sociocultural behavior in different geographic regions, the results of the present study may not be universal in all geographic societies. Third, because we enrolled patients with pathologically confirmed OPSP after complete surgical resection, the natural course of HPV-associated OPSP could not be followed. Thus, it is still questionable whether current HPV infection in OPSP only acts as a causative factor for the development of OPSP or can progress to HPV-driven carcinogenesis. To overcome these limitations of the present study, a prospective case–control study involving a much larger number of patients with OPSP from various geographical regions should be performed.

## 5. Conclusions

This study showed that the prevalence of overall HPV and high-risk HPV infections in OPSP were 14.5% and 10.8%, respectively. The most prevalent genotype was HPV16, accounting for 58.3% of all HPV infections. There were trends toward a higher prevalence of HR-HPV infection in patients with OPSP aged ≤45 years, never-smokers, and multifocal diseases. The findings of the present study could enhance our understanding of HPV infection in OPSP and could be used as valuable epidemiological data for the management of HPV-associated OPSP, expecting the possible efficacy of HPV vaccinations in OPSP.

## Figures and Tables

**Figure 1 cancers-15-00810-f001:**
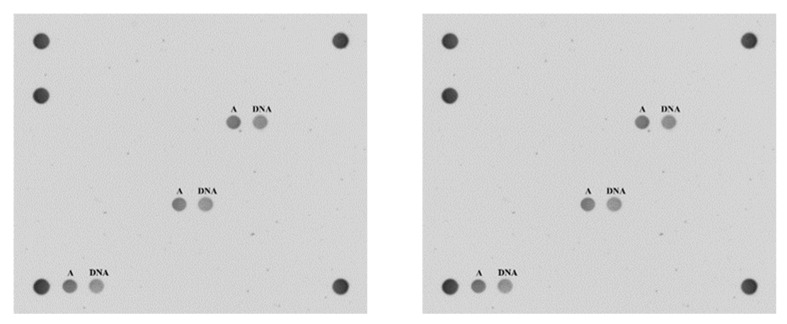
Both the amplification control (A) and genomic DNA control (DNA) are present in the sample of HPV-negative oropharyngeal squamous papilloma (**left**). In the sample of HPV-positive oropharyngeal squamous papilloma, type-specific amplification, HPV16 in this case, is detected via hybridization as well as the two internal controls (**right**).

**Figure 2 cancers-15-00810-f002:**
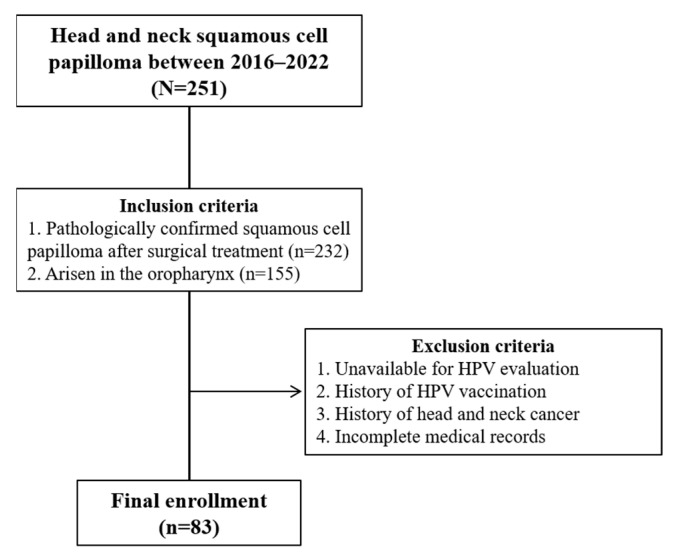
Patient enrollment flow chart.

**Table 1 cancers-15-00810-t001:** Baseline patient characteristics.

	Patients (N = 83)
Age (years)	51.0 ± 16.7
Sex	
Men	54 (65.1%)
Women	29 (34.9%)
Smoking status	
Never-smoker	43 (51.8%)
Ex-smoker	12 (14.5%)
Current smoker	28 (33.7%)
Clinical presentation *	
None (incidentally found)	60 (72.3%)
By self-examination	14 (16.9%)
By dental examination	1 (1.2%)
By endoscopic examination	45 (54.2%)
Foreign body sensation	21 (25.3%)
Chronic cough	2 (2.4%)
Subsites ^†^	
Tonsil	36 (43.4%)
Tongue base	18 (21.7%)
Soft palate	28 (33.7%)
Posterior wall	6 (7.2%)
Focality	
Unifocal	79 (95.2%)
Multifocal	4 (4.8%)
Dysplasia on pathological examination	
No	83 (100.0%)
Yes	0 (0.0%)
Recurrence	
No	0 (0.0%)
Yes	83 (100.0%)

* Multiple presentations have been included. ^†^ Multiple subsites were included in cases of multifocal OPSP.

**Table 2 cancers-15-00810-t002:** Overall prevalence and genotypes of HPV infection in oropharyngeal squamous papilloma.

	Patients (N = 83)
HPV infection	
Negative	71 (85.5%)
Positive	12 (14.5%)
High-risk	9 (10.8%)
16	7
58	1
39 and 66	1
Low-risk	3 (3.6%)
11	2
84	1
Immunohistochemistry for p16	
Negative	12 (100.0%)
Positive	0 (0.0%)

HPV, human papillomavirus.

**Table 3 cancers-15-00810-t003:** Detailed characteristics of 12 patients with HPV-positive oropharyngeal squamous cell papilloma.

Patient Number	Sex	Age	Smoking	Clinical Presentation	Subsite	Focality	HPV Genotype	Genotype Category
1	M	38	Ex	None	Tonsil	Unifocal	16	High
2	F	54	Never	Voice change	Soft palate	Unifocal	16	High
3	M	19	Never	None	Soft palate	Unifocal	16	High
4	F	62	Never	None	Soft palate	Unifocal	84	Low
5	F	68	Never	Foreign body sensation	Tongue base	Unifocal	16	High
6	M	41	Current	None	Tonsil	Unifocal	16	High
7	M	61	Current	Foreign body sensation	Tongue base	Multifocal	58	High
8	M	57	Current	None	Tonsil	Unifocal	11	Low
9	M	61	Never	Foreign body sensation	Tonsil	Unifocal	16	High
10	M	60	Current	None	Posterior wall	Unifocal	11	low
11	F	20	Never	None	Tonsil	Unifocal	39, 66	High
12	F	47	Never	None	Posterior wall	Unifocal	16	High

HPV, human papillomavirus.

**Table 4 cancers-15-00810-t004:** Prevalence of HPV infection according to clinicodemographic profiles.

	Age			Sex			Smoking			Focality			Subsite				
	≤45 (*n* = 27)	>45 (*n* = 56)	*p*-Value	Men (*n* = 54)	Women (*n* = 29)	*p*-Value	Never (*n* = 43)	Ex- or Current (*n* = 40)	*p*-Value	Unifocal (*n* = 79)	Multifocal (*n* = 4)	*p*-Value	Tonsil (*n* = 36)	Tongue Base (*n* = 18)	Soft Palate (*n* = 28)	Posterior Wall (*n* = 6)	*p*-Value
Overall HPV (+)	4 (14.8%)	8 (14.3%)	0.595	7 (13.0%)	5 (17.2%)	0.412	7 (16.3%)	5 (12.5%)	0.431	11 (13.9%)	1 (25.0%)	0.471	5 (13.9%)	2 (11.1%)	3 (10.7%)	2 (33.3%)	0.517
Overall HPV (−)	23 (85.2%)	48 (85.7%)		47 (87.0%)	24 (82.8%)		36 (83.7%)	35 (87.5%)		68 (86.1%)	3 (75.0%)		31 (86.1%)	16 (88.9%)	25 (89.3%)	4 (66.7%)	
HR-HPV (+)	4 (14.8%)	5 (8.9%)	0.324	5 (9.3%)	4 (13.8%)	0.386	6 (14.0%)	3 (7.5%)	0.279	8 (10.1%)	1 (25.0%)	0.374	4 (11.1%)	2 (11.1%)	2 (7.1%)	1 (16.7%)	0.895
HR-HPV (−)	23 (85.2%)	51 (91.1%)		49 (90.7%)	25 (86.2%)		37 (86.0%)	37 (92.5%)		71 (89.9%)	3 (75.0%)		32 (88.9%)	16 (88.9%)	26 (92.9%)	5 (83.3%)	
Proportion of HR types in all HPV infections	100.0%	62.5%	0.255	71.4%	80.0%	0.636	85.7%	60.0%	0.364	72.7%	100.0%	0.750	80.0%	100.0%	66.7%	50.0%	0.680

HPV, human papillomavirus; HR, high-risk.

## Data Availability

The data that support the findings of this study are available on request from the corresponding author (Dongbin Ahn).

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
