# Peer review of "Prevalence and Characteristics of Human Papillomavirus Infection in Oropharyngeal Squamous Cell Papilloma"

_cancers, 2023, doi:10.3390/cancers15030810_

Round 1
Reviewer 1 Report
In this study authors collected the prevalence and characteristics of HPV infection in oropharyngeal squamous cell papilloma (OPSP) and found a 14.5% overall prevalence. High-risk HPV accounted for 75% of all HPV infections, with HPV16 being the most prevalent genotype, accounting for 58.3% of all HPV infections.
Comments
An important and clinically valuable study.
Comments
Introduction
I can imagine continent, region and country specific differences in the prevalence of the OPSP and its viral background. Please mention in the Introduction in all cases where were the "previous studies" performed. In the Discussion this comparison would be also suggested.
Material and Methods
Page 3, line 99, please mention with which antibody or assay kit the immunohistochemistry (IHC) staining for the p16 protein war performed on which immunostainer.
Statistical analysis
Please mention if the effect of the input variables as clinicodemographic characteristics, such as age, sex, smoking status, tumor focality, and anatomical subsites of the oropharynx, on the output variables as prevalence, HPV genotype, dysplasia, p16 positivity and recurrence was analysed by statistical models or multivariate analysis.
Results
Table 2.
Immunohistochemistry for p16, the table contains 12 cases stained for p16, all of them were p16 IHC-negative. No staining was performed in the HPV-negative cases? Would it make sense to stain also the HPV-negative cases? p16 might be stained independently from the HPV-background, which happens in some oropharynx SCC. A comparison could be interesting in this case.
Discussion
Page 7, lines 229-232
"p16 overexpression, which acts as a surrogate marker of the presence of biologically active HPV, was not found even in high-risk HPV-associated OPSP, suggesting that none of them had transcriptionally active HPV infection that could lead to HPV-driven carcinogenesis."
Does this mean that HPV+ OPSP is not a pre-stage of oropharynx cancer, and a may be a different patient tissue background decides if high risk HPV induces papilloma or tumor?
Author Response
Prof. Dr. Samuel C. Mok
Editor-in-Chief
Cancers
Dear Dr. Mok:
We wish to resubmit our manuscript titled “Prevalence and characteristics of human papillomavirus infection in oropharyngeal squamous cell papilloma.” The manuscript ID is cancers-2162839.
We thank the editors and reviewers for their meticulous review of our manuscript. We have revised our manuscript according to the comments and suggestions provided by the editors and reviewers. The point-by-point responses to the comments are appended below and the revisions are denoted in red font in the revised manuscript. In addition, the revised manuscript has been edited by a professional English editing company to improve the language, flow, and readability.
We hope that the revised manuscript is now suitable for publication in your esteemed journal. Thank you for your consideration. I look forward to hearing from you.
Sincerely,
Dongbin Ahn, MD
Department of Otolaryngology-Head and Neck Surgery, Kyungpook National University, 130 Dongdeok-ro, Jung-gu, Daegu 41944, Korea
Tel.: +82-53-200-5781
Fax: +82-53-423-4524
E-mail address:godlikeu@naver.com
Review report- Reviewer 1
Point 1. Introduction. I can imagine continent, region and country specific differences in the prevalence of the OPSP and its viral background. Please mention in the Introduction in all cases where were the "previous studies" performed. In the Discussion this comparison would be also suggested.
Response. We thank the reviewer for the insightful comments. As we described in the discussion, it may not be possible to generalize the results of the present study to all geographic regions, because the prevalence and characteristics of HPV infection are highly affected by sexual and sociocultural behavior in different geographic regions. Thus, providing different geographical data of HPV infection in OPSP would be valuable for the comprehensive understanding of HPV infection in OPSP.
We performed a literature review to provide geographical data on HPV infection in OPSP, which we used to revise the Introduction section. The information has been compared with our results in the Discussion section of the revised manuscript (Lines 60–62; 193–195).
Point 2. Material and Methods. Page 3, line 99, please mention with which antibody or assay kit the immunohistochemistry (IHC) staining for the p16 protein was performed on which immunostainer.
Response. We have provided specific information on the IHC staining for the p16 protein in the Material and Methods section of the revised manuscript (Lines 100–102).
Point 3. Statistical analysis. Please mention if the effect of the input variables as clinicodemographic characteristics, such as age, sex, smoking status, tumor focality, and anatomical subsites of the oropharynx, on the output variables as prevalence, HPV genotype, dysplasia, p16 positivity and recurrence was analysed by statistical models or multivariate analysis.
Response. We thank the reviewer for the comments. In accordance with reviewer’s suggestions, we performed uni- and multivariate analyses for the association between several clinicodemographic characteristics (age, sex, smoking status, tumor focality, and anatomical subsites of the oropharynx) and prevalence of overall/high-risk HPV infections in OPSP. We did not find any statistically significant associations between variables, possibly due to the small number of HPV-positive cases. Indeed, Reviewer 2 pointed out that the demographic statistics are not convincing due to the limited number of HPV-positive cases. Thus, we have not included the results of these analyses in the revised manuscript. We would appreciate your understanding regarding this matter.
Point 4. Results. Table 2. Immunohistochemistry for p16, the table contains 12 cases stained for p16, all of them were p16 IHC-negative. No staining was performed in the HPV-negative cases? Would it make sense to stain also the HPV-negative cases? p16 might be stained independently from the HPV-background, which happens in some oropharynx SCC. A comparison could be interesting in this case.
Response. We thank the reviewer for the comments. As we mentioned in the manuscript, immunohistochemistry (IHC) staining for the p16 protein was performed in all OPSPs (Lines 100–102). However, none of the 83 OPSPs was positive for p16 IHC, regardless of the HPV infection status, when it was defined as homogeneous, strong nuclear, and cytoplasmic staining present in >70% of tumor cells. We have added this information in the Results section of the revised manuscript (Lines 149–150).
Point 5. Discussion. Page 7, lines 229-232. "p16 overexpression, which acts as a surrogate marker of the presence of biologically active HPV, was not found even in high-risk HPV-associated OPSP, suggesting that none of them had transcriptionally active HPV infection that could lead to HPV-driven carcinogenesis." Does this mean that HPV+ OPSP is not a pre-stage of oropharynx cancer, and a may be a different patient tissue background decides if high risk HPV induces papilloma or tumor?
Response. We thank the reviewer for raising this relevant point. It is well documented that high-risk HPV infection does not always induce carcinogenesis; however, it could progress towards several different biological and clinical pathways. HPV infection would be cleared spontaneously in most patients (Lancet. 2013 September 7; 382(9895): 877–887), may be associated with the development of benign tumor in some patients (HPV-associated OPSP and genital warts), or may induce HPV-driven carcinogenesis in few patients (HPV-associated cervical cancer and oropharyngeal cancer). Indeed, HPV infection is only the first step in the HPV-driven carcinogenesis, and many additional genetic and signaling alterations, involving p53 and Rb degradation by E6 and E7, respectively, and PI3K/AKT/mTOR signaling, are required for HPV infection to ultimately lead to HPV-associated cancer. Therefore, HPV-associated OPSP with negative p16 was considered another phenotype of oropharyngeal HPV infection that might present with a different pathogenesis from that of HPV-driven carcinogenesis.
We have added this description in the Discussion section of the revised manuscript (Lines 241–253).
Reviewer 2 Report
Overall this is a very interesting clinical study with some clinical importance on the etiologic knowledge of oropharyngeal papilloma - a very common benign entity with unclear reasons. Some concerns are raised below.
1. what's the implications for HPV infection in oropharyngeal papilloma? especially its clinical implications in guiding treatment or prognosis?
2. prognostic data are missing
3. due to limited HPV-positive cases, the demographic statistics are not convincing
4. figures should be presented for the HPV detection including genotyping and IHC, also positive and negative controls
5. how about EBV infection in oropharyngeal papilloma
6. why patients with a history of HPV vaccination were excluded?
7. why were patients "whose HPV status was not evaluated" exluded? this study is supposed to detect the HPV status
8. pls explain the discrimination between HPV PCR and IHC results
Author Response
Prof. Dr. Samuel C. Mok
Editor-in-Chief
Cancers
Dear Dr. Mok:
We wish to resubmit our manuscript titled “Prevalence and characteristics of human papillomavirus infection in oropharyngeal squamous cell papilloma.” The manuscript ID is cancers-2162839.
We thank the editors and reviewers for their meticulous review of our manuscript. We have revised our manuscript according to the comments and suggestions provided by the editors and reviewers. The point-by-point responses to the comments are appended below and the revisions are denoted in red font in the revised manuscript. In addition, the revised manuscript has been edited by a professional English editing company to improve the language, flow, and readability.
We hope that the revised manuscript is now suitable for publication in your esteemed journal. Thank you for your consideration. I look forward to hearing from you.
Sincerely,
Dongbin Ahn, MD
Department of Otolaryngology-Head and Neck Surgery, Kyungpook National University, 130 Dongdeok-ro, Jung-gu, Daegu 41944, Korea
Tel.: +82-53-200-5781
Fax: +82-53-423-4524
E-mail address:godlikeu@naver.com
Review report- Reviewer 2
Point 1. What's the implications for HPV infection in oropharyngeal papilloma? especially its clinical implications in guiding treatment or prognosis?
Response. We thank the reviewer for the insightful comments. In this study, we found that high-risk HPV infection accounted for 75% of all HPV infections with HPV16 being the most prevalent genotype. In addition, there were trends toward a higher prevalence of high-risk HPV infection in patients with OPSP aged ≤45 years, never-smokers, and multifocal diseases, which were consistent with the clinicodemographic profiles of HPV-associated oropharyngeal cancer. Therefore, we believe that complete surgical resection would be a better therapeutic approach than observation without treatment to obtain specific histological and virological information of OPSP and to eliminate the possible risk of disease progression or malignant transformation. We have described implication of HPV infection in OPSP in the Discussion section (Lines 253–267).
Point 2. Prognostic data are missing.
Response. We thank the reviewer for pointing this out. None of the patients showed recurrence after surgical treatment, regardless of the HPV infection status. We have added this information in the Result section and Table 1 in the revised manuscript (Line 130).
Point 3. Due to limited HPV-positive cases, the demographic statistics are not convincing.
Response. We agree with the reviewer’s point. Although this is the largest study on HPV infection in OPSP, the number of the study subjects was still small to demonstrate statistical differences in the prevalence of the overall/high-risk HPV infection, as a function of the various clinicodemographic characteristics. We have described this issue as a limitation of the present study (Lines 289–290).
Point 4. Figures should be presented for the HPV detection including genotyping and IHC, also positive and negative controls.
Response. We thank the reviewer for the comments. We have added an array tube image of a CLARTH HPV 2 assay for HPV-negative and HPV-positive OPSP samples in Figure 2 in the revised manuscript. In terms of the p16 IHC, the result was negative in all 83 patients regardless of the HPV infection status; thus, we did not include the figures of the p16 IHC in the revised manuscript.
Point 5. How about EBV infection in oropharyngeal papilloma.
Response. We thank the reviewer for the insightful comments. To the best of our knowledge, there are no studies on EBV infection in OPSP. However, EBV infection in oropharyngeal papilloma would be a novel issue considering that there are several studies evaluating the role of EBV in other head and neck papillomas including sinonsasl inverted papilloma and recurrent respiratory papillomatosis. Unfortunately, our study could not address this issue, either. We appreciate your understanding regarding this matter.
Point 6. Why patients with a history of HPV vaccination were excluded?
Response. The aim of this study was to evaluate the prevalence and characteristics of HPV infection in OPSP and we considered that the effect of HPV vaccination should be excluded to identify the true prevalence of HPV infection in OPSP from the general population. Indeed, it would be very interesting if the prevalence and genotypes of HPV infection would be compared between vaccinated and unvaccinated OPSP patients. Such a comparison study would require a much larger study population including both vaccinated and unvaccinated OPSP patients. However, given that the mean age of patients with OPSP ranges from 40 to 60 years reportedly, and since the current HPV vaccination program is primarily targeting adolescents and young women, such a study may not be feasible currently or in the near future. We have briefly mentioned the possible effect of HPV vaccination on OPSP in the Discussion section of the revised manuscript (Lines 283–286).
Point 7. why were patients "whose HPV status was not evaluated" excluded? this study is supposed to detect the HPV status.
Response. We apologize for the misunderstanding. The intended meaning was exclusion of OPSP patients in whom evaluation of HPV status was unavailable. We have corrected this in the revised manuscript (Line 81, Figure 1).
Point 8. pls explain the discrimination between HPV PCR and IHC results.
Response. In this study, p16 overexpression was not detected even in high-risk HPV-associated OPSP, suggesting that none of the patients had transcriptionally active HPV infection that could have led to HPV-driven carcinogenesis. It is well documented that high-risk HPV infection does not always induce carcinogenesis; however, it could progress toward several different biological and clinical pathways. HPV infection would be cleared spontaneously in majority of patients (Lancet. 2013 September 7; 382(9895): 877–887), may be associated with the development of benign tumor in some patients (HPV-associated OPSP and genital warts), or may induce HPV-driven carcinogenesis in few patients (HPV-associated cervical cancer and oropharyngeal cancer). Indeed, HPV infection is only the first step in the HPV-driven carcinogenesis, and many additional genetic and signaling alterations, involving p53 and Rb degradation by E6 and E7, respectively, and PI3K/AKT/mTOR signaling, are required for HPV infection to ultimately lead to HPV-associated cancer. Therefore, HPV-associated OPSP with negative p16 was considered another phenotype of oropharyngeal HPV infection that might present with a different pathogenesis from that of HPV-driven carcinogenesis. We have added this description in the Discussion section of the revised manuscript (Lines 241–253).
Round 2
Reviewer 2 Report
All major concerns have been satisfactorily addressed. One minor point is raised below.
In the reply letter, the authors stated “we believe that complete surgical resection would be a better therapeutic approach than observation without treatment to obtain specific histological and virological information of OPSP and to eliminate the possible risk of disease progression or malignant transformation”. However, this statement is not supported by their findings especially considering the benign feature of papilloma and none of the patients had recurrence in the cohort.
Author Response
Response
Point 1. One minor point is raised below. In the reply letter, the authors stated “we believe that complete surgical resection would be a better therapeutic approach than observation without treatment to obtain specific histological and virological information of OPSP and to eliminate the possible risk of disease progression or malignant transformation”. However, this statement is not supported by their findings especially considering the benign feature of papilloma and none of the patients had recurrence in the cohort.P
Response. We thank you for the comments and understand your concerns. We have modified that sentence as follows to have more modest meaning in the revised manuscript: “we consider complete surgical resection of OPSP a better therapeutic approach than observation without treatment to obtain its specific histological and virological information”